# Comparison of the Effectiveness of Palonosetron and Ramosetron in Preventing Postoperative Nausea and Vomiting: Updated Systematic Review and Meta-Analysis with Trial Sequential Analysis

**DOI:** 10.3390/jpm13010082

**Published:** 2022-12-29

**Authors:** Hyo Jin Kim, EunJin Ahn, Geun Joo Choi, Hyun Kang

**Affiliations:** 1Department of Anesthesiology and Pain Medicine, Chung-Ang University Gwangmyeong Hospital, Gwangmyeong-si 14353, Republic of Korea; 2Department of Anesthesiology and Pain Medicine, Chung-Ang University College of Medicine, Seoul 06911, Republic of Korea

**Keywords:** palonosetron, ramosetron, PONV

## Abstract

This updated systematic review and meta-analysis with trial sequential analysis aimed to compare the efficacy of the perioperative administration of palonosetron with that of ramosetron in preventing postoperative nausea and vomiting (PONV). A total of 17 randomized controlled trials comparing the efficacy of the perioperative administration of palonosetron to that of ramosetron for preventing PONV were included. The primary outcomes were the incidences of postoperative nausea (PON), postoperative vomiting (POV), and PONV, which were measured in early, late, and overall phases. Subgroup analysis was performed on the basis of the administration time of the 5-HT3 receptor antagonist and divided into two phases: early phase and the end of surgery. A total of 17 studies with 1823 patients were included in the final analysis. The incidence of retching (relative risk [RR] = 0.525; 95% confidence interval [CI] = 0.390 to 0.707) and late POV (RR = 0.604; 95% CI = 0.404 to 0.903) was significantly lower in the palonosetron group than in the ramosetron group. No significant differences were demonstrated in the incidence of PON, PONV, complete response, use of antiemetics, and adverse effects. Subgroup analysis showed that palonosetron was superior to ramosetron in terms of early PON, late PON, overall POV, and use of rescue antiemetics when they were administered early; in terms of retching, regardless of the timing of administration. Ramosetron was superior to palonosetron in terms of early PON when they were administered late. The prophylactic administration of palonosetron was more effective than that of ramosetron in preventing the development of retching and late POV. In this meta-analysis, no significant differences in PONV prevention between the two drugs were demonstrated. Further studies are required to validate the outcomes of our study.

## 1. Introduction

Postoperative nausea and vomiting (PONV) is a common and distressing complication following anesthesia and surgery, with a reported incidence of up to 80% [1]. PONV is associated not only with greater patient discomfort but also with an increase in medical costs because of unexpected complications [2]. Its etiology is multifactorial, and various antiemetics have been studied for its prevention and treatment. Serotonin type 3 (5-HT3), opioid, muscarinic, and dopamine type 2 (D2) receptors play roles in emesis [3]. Among various antiemetics, 5-HT3 receptor antagonists are widely used as the first line regimen in the prophylaxis and treatment of PONV [4]. Ramosetron binds more strongly to the 5-HT3 receptor and has an extended duration of action compared to other conventional 5-HT3 receptor antagonists [5]. Palonosetron has the advantages of greater receptor selectivity and significantly longer half-life compared to other 5-HT3 receptor antagonists [6].

Although several systematic reviews and meta-analyses have compared the efficacy of palonosetron and ramosetron [6,7,8,9], the findings of these studies are conflicting and variable. A systematic review and meta-analysis reported that palonosetron is superior to ramosetron in preventing delayed PONV [6]. On the other hand, subsequent meta-analyses reported that no definite differences were found between palonosetron and ramosetron in the effectiveness of preventing PONV [7,8,9].

Our previous study reported that palonosetron is more effective than ramosetron when it is administered at the early phase of surgery, while ramosetron is more effective than palonosetron when it is administered at the end of surgery [7]. It includes only a small number of cases; consequently, the quality of the evidence is limited. Additional large-scale high-quality randomized controlled trials (RCTs) investigating the effectiveness of palonosetron and ramosetron for preventing PONV have been published; therefore, comprehensive data can be analyzed.

This updated systematic review and meta-analysis with trial sequential analysis was performed to compare the efficacy of the perioperative administration of palonosetron with that of ramosetron in preventing PONV.

## 2. Materials and Methods

The protocol of the updated systematic review and meta-analysis with trial sequential analysis was developed according to the Preferred Reporting Items for Systematic Review and Meta-Analysis Protocol (PRISMA-P) and registered in the International Prospective Register of Systematic Reviews (PROSPERO; registration number: CRD42022291040). This updated systematic review and meta-analysis with trial sequential analysis was conducted in accordance with the PRISMA statement guidelines [10].

### 2.1. Inclusion and Exclusion Criteria

The inclusion and exclusion criteria for this study were determined before this updated systematic review and meta-analysis with trial sequential analysis was conducted.

Randomized controlled trials (RCTs) that compared the efficacy of palonosetron with that of ramosetron on PONV prophylaxis were included. The study details are as follows:Patients (P): All patients receiving elective surgery under general anesthesia.Intervention (I): Administration of palonosetron.Comparison (C): Administration of ramosetron.Outcome measurements (O): The primary outcomes of this updated systematic review and meta-analysis with trial sequential analysis were the incidences of postoperative nausea (PON), postoperative vomiting (POV), and PONV, which were measured in early, late, and overall phases. The secondary outcomes were the incidence of retching, use of rescue antiemetic drugs, complete response, and adverse events (dizziness and headache). For the primary outcomes, the postoperative period was classified into three periods: early, late, and overall phases. The early phase was defined as 0 to 24 h after surgery, and the late phase was defined as 24 to 48 h after surgery. Data from the initial time point were chosen as the outcome of interest in case of studies which reported data from various time points within the same phase. For instance, the data at 0 h was chosen as the outcome of early phase if the study reported data at 0, 2, 4, 6, and 24 h after surgery. To capture the maximum number of studies, all PON, POV, and PONV data from studies that did not mention a specific time point were defined as outcome data in the overall phase.Study design (SD): RCTs.

Data from case reports, editorials or letters to the editor, reviews, and animal or laboratory studies were excluded.

### 2.2. Systematic Search

A comprehensive and systematic search for RCTs that compared the efficacy of palonosetron with that of ramosetron in preventing PONV was conducted by two independent investigators (AEJ and CGJ). Studies involving the use of single antiemetic were included. Ovid-MEDLINE, Ovid-EMBASE, the Cochrane Central Register of Controlled Trials (CENTRAL), Web of Science, and Google Scholar were searched for all relevant articles published from in-session by January 30, 2022 (inclusive).

Search terms were devised in collaboration with a medical librarian and included a combination of free text, Medical Subject Headings, and EMTREE terms for “palonosetron,” “ramosetron,” and “randomized controlled trial.” Clinical trial registries were implemented to identify completed but unpublished RCTs from registered trials and Open SIGLE was also implemented to search the gray literature. In addition, manual searches were conducted in the reference lists of the entire articles that were retrieved. The search strategy is described in the Appendix B.

### 2.3. Study Selection

The schematic flow diagram of the study selection was presented in Figure 1. Two separate investigators (KHJ and AEJ) scanned the titles and abstracts of the reports identified via the search algorithms previously outlined. The full article was retrieved if the report was deemed eligible based on its titles and abstracts. The entire texts of studies which at least one investigator had selected as potentially pertinent were obtained and evaluated. Two investigators (KHJ and AEJ) independently evaluated articles that met the requirement for inclusion, and any disagreements were settled through discussion. If consensus could not be achieved, a third investigator (KH) was recruited to resolve the dispute.

### 2.4. Data Extraction

Using a predefined, standardized data collection form, two researchers (AEJ and CGJ) independently retrieved all interrelated data from the included studies and then cross-checked findings. In case of disagreements, a consensus was reached through discussion between the two investigators. If consensus could not be achieved, a third investigator (KH) was recruited to resolve the dispute. The spreadsheet for data extraction included the following items: (1) title; (2) name of first author; (3) journal name; (4) year of publication; (5) design of study; (6) clinical trial registration; (7) conflicts of interests; (8) country; (9) risk of bias; (10) number of patients included; (11) dosage of palonosetron and ramosetron; (12) sex; (13) age; (14) height of patients; (15) weight of patients; (16) duration of anesthesia; (17) physical status according to the American Society of Anesthesiologists (ASA) classification; (18) inclusion criteria; (19) exclusion criteria; (20) type of anesthesia; (21) type of surgery; (22) agent of anesthesia induction; (23) agent of anesthesia maintenance; (24) use of nitrous oxide (N_2_O); (25) perioperative use of an opioid; (26) administration time of the experimental drug (either palonosetron or ramosetron); (27) other medications administered during surgery; (28) rescue analgesics; (29) use of rescue antiemetics; (30) definitions of nausea, vomiting, and retching; (31) number of cases of PON, POV, and PONV during the early, late postoperative phases and overall phase; and (32) the requirement for rescue antiemetics, (33) retching, and (34) complete response.

Data were initially extracted from tables, figures, or text. The study authors were contacted via email to obtain pertinent information in the case of missing or insufficient data.

### 2.5. Data Analysis

#### 2.5.1. Conventional Meta-Analysis

Meta-analysis was conducted using Comprehensive Meta-Analysis version 2.0 (Englewood, NJ, USA, 2008). All data were separately inputted by two investigators (AEJ and CGJ) into the software and then cross-checked. The pooled risk ratio (RR) and their 95% confidence intervals (CIs) for each outcome were calculated. An I^2^ test was used for heterogeneity. I^2^ of >50% was regarded as indicating considerable heterogeneity. If I^2^ was <50%, a fixed-effect model was applied. When I^2^ > 50%, a random effect model was applied [11,12].

In order to explore heterogeneity, sensitivity analysis was conducted to determine whether our results were altered by omitting one study at a time [13].

When the number of the pooled studies demonstrating substantial heterogeneity was <10, the Z-test was substituted with the t-statistics (Hartung–Knapp–Sidik–Jonkman method) in the analysis of all random effects to minimize the error rate [14].

Publication bias was assessed by Begg’s funnel plot and Egger’s linear regression test [12]. The presence of a publication bias was deemed, and trim and fill analysis was conducted if the funnel plot was visually asymmetrical, or Egger’s test revealed the *p*-value of less than 0.10.

To estimate the overall clinical impact of the intervention, the number needed to treat (NNT) was computed using 95% CI based on the absolute risk reduction [15].

#### 2.5.2. Trial Sequential Analysis

Trial sequential analysis (TSA) was conducted to compute the required information size (RIS) and assess whether our results were conclusive [16]. A fixed or DerSimonian and Laird approach (*DL*) random effect model was used to establish the cumulative Z-curve. TSA was conducted to maintain an overall risk of type I error at 5%. If the cumulative Z-curve crossed the trial sequential monitoring boundary or entered within the futility area (area of no benefit), a sufficient level of evidence might have been reached to accept or reject the anticipated intervention effect, and no further studies were required. When the Z-curve did not cross any boundaries, and the RIS was not obtained, it means that evidence was insufficient to draw a conclusion, indicating that further studies should be performed.

The RIS was estimated on the basis of the observed proportion of patients who developed outcomes in the ramosetron group (the cumulative proportion of patients who experienced an event relative to all patients in the ramosetron groups), 20% relative risk reduction in the palonosetron group, an alpha of 5% for all our outcomes, a beta of 20%, and the monitored diversity as indicated by the trials in TSA.

#### 2.5.3. Validity Scoring

Using the revised Cochrane risk of bias tool for randomized trials (RoB 2.0 version), the quality of each study was critically appraised by two separate investigators (AEJ and KHJ) [17]. Primarily, each domain of the studies was evaluated and rated as follows: (D1) bias derived from the randomization process, (D2) bias because of deviations from the intended interventions, (D3) bias resulting from missing outcome data, (D4) bias due to measurement of the outcome, and (D5) bias due to the selection of the reported result. Then, the overall risk of bias was assessed as follows: (1) low risk, which occurs when the risk of bias for all domains was low; (2) high risk, which occurs when the risk of bias for at least one domain was high or the risk of bias for multiple domains was of some concern; and (3) some concern, which occurs when the overall judgment was neither low nor high. When consensus could not be established, disputes were resolved by a discussion with a third investigator (KH).

#### 2.5.4. Quality of Evidence

The guidelines of the Grading of Recommendations, Assessment, Development, and Evaluation (GRADE) system were applied to evaluate the quality of evidence. The guidelines involve the sequential assessment of the evidence quality, evaluation of the risk–benefit balance, and subsequent appraisal on the strength of the recommendations [18]. The following are the four categories into which the quality of evidence is graded: (1) high indicated that the confidence in the effect estimate was unlikely to alter with further research; (2) moderate indicated that additional research was likely to significantly change confidence in the effect estimate and might alter the estimate; (3) low indicated that additional study was likely to significantly change confidence in the effect estimate and alter the estimate; and (4) very low indicated that no effect estimate was certain.

## 3. Results

### 3.1. Literature Search and Study Selection

A search of all relevant articles published from in-session through January 30, 2022 in Ovid-MEDLINE, Ovid-EMBASE, CENTRAL, Web of Science, and Google Scholar yielded a total of 276 articles, and 33 additional articles were identified after a manual search was conducted. After duplicates (*n* = 21) were removed, 288 studies were retained. Of these studies, 253 were excluded after reviewing the titles and abstracts because they were judged not to be relevant. The kappa value for literature selection between two investigators at this stage of study selection was 0.754.

The full text of the 25 remaining articles was reviewed in further detail, and eight more articles were excluded for the following reasons: systematic review and meta-analysis, [8,19,20] no outcome of interest [21], performed for chemotherapy-induced nausea and vomiting [22,23,24], and not a RCT [25]. The kappa value for selecting articles between the two investigators was 0.930.

Lastly, the final systematic review and meta-analysis included 17 studies with a total of 1823 patients (Figure 2).

### 3.2. Study Characteristics

The characteristics of the 17 studies that were eligible for the inclusion criteria are described in Table 1 and Table 2. All studies except one study performed for elective cesarean section under spinal anesthesia [26] were conducted under general anesthesia. One study was a conference proceeding [27]. Types of surgeries in the included trials were abdominal surgeries [26,28,29,30,31,32,33,34,35,36,37,38,39], spinal surgeries [40,41], and ear, nose, and throat surgeries [27,39,42]. All studies used the same dose of palonosetron (0.075 mg) and ramosetron (0.3 mg). The time of 5-HT3 receptor antagonist administration was divided into three categories: early phase of surgery [26,28,31,33,36,37,40], late phase of surgery [27,29,30,32,35,39,41,42], and each drug administered at different times [34,38]. The time of outcome data collection was sorted into the following three phases: (1) early (0 to 24 h after surgery), (2) late (24 to 48 h after surgery), and (3) overall phase.

### 3.3. PON (Early, Late, and Overall Phase)

A total of 12 studies (1263 patients) reported the incidence of early PON [26,27,28,30,31,33,35,36,37,38,40,42].

The combined results showed no evidence of a difference (RR = 1.048; 95% = CI 0.811 to 1.354; I^2^ = 46.34; τ^2^ = 0.18; NNT harm [NNTH] = 741; 95% CI NNTH 25 to ∞ to NNT benefit [NNTB] = 26) between the palonosetron (14.9%; 94 of 632 patients) and ramosetron (14.7%; 93 of 631 patients) groups.

For subgroup analysis, the incidence of early PON was significantly lower in the palonosetron group than in the ramosetron group when these drugs were administered early (RR = 0.575; 95% CI = 0.359 to 0.922; I^2^ = 16.15; τ^2^ = 0.06), but the incidence of early PON was significantly lower in the ramosetron group than in the palonosetron group when they were administered late (RR = 1.377; 95% CI = 1.010 to 1.878; I^2^ = 0.0; τ^2^ = 0.00; Figure 3A).

Only 9.5% (1263 of 13,314 patients) of the RIS was accrued in the TSA. The cumulative Z curve did not cross the conventional test boundary (Appendix A).

Twelve studies (1489 patients) reported the incidence of late PON [26,28,30,31,33,35,36,37,38,40,41,42].

The combined results showed no evidence of a difference (RR = 1.033; 95% CI = 0.753 to 1.417; I^2^ = 57.93; τ^2^ = 0.20; NNTB = 1911; 95% CI = NNTH 25 to ∞ to NNTB 24) between the palonosetron (20.3%; 151 of 743 patients) and ramosetron (20.4%; 152 of 746 patients) groups.

For subgroup analysis, the incidence of late PON was significantly lower in the palonosetron group than in the ramosetron group when they were administered early (RR = 0.454; 95% CI = 0.222 to 0.928; I^2^ = 55.26; τ^2^ = 0.39), but there was no evidence of difference between the palonosetron and ramosetron group when they were administered late (RR = 1.270; 95% CI = 0.877 to 1.839; I^2^ = 39.35; τ^2^ = 0.06; Figure 3B).

TSA indicated that only 10.1% (1489 of 14,733 patients) of the RIS was accrued. The cumulative Z curve did not cross conventional test boundary (Appendix A).

A total of 13 studies (1323 patients) reported the incidence of overall PON [26,27,28,30,31,33,35,36,37,38,39,40,42]. The combined results showed no evidence of a difference (RR = 1.045; 95% CI 0.893 to 1.222; I^2^ = 48.75; τ^2^ = 0.10; NNTH = 68; 95% CI = NNTH 16 to ∞ to NNTB 31) between the palonosetron (25.7%; 170 of 662 patients) and ramosetron (24.2%; 160 of 661 patients) groups. For subgroup analysis, no evidence of differences was observed when they were administered early (RR = 0.878; 95% CI = 0.688 to 1.120; I^2^ = 0.0; τ^2^ = 0.0) and late (RR = 1.208; 95% CI = 0.893 to 1.624; I^2^ = 0.0; τ^2^ = 0.0; Figure 3C).

Only 15.2% (1323 of 8707 patients) of the RIS was accrued in the TSA. The cumulative Z curve did not cross the conventional test boundary (Appendix A).

### 3.4. POV (Early, Late, and Overall Phase)

Ten studies (1097 patients) reported the incidence of early POV [26,30,31,32,33,35,37,38,40,42]. The combined results showed no evidence of a difference (RR = 0.734; 95% CI = 0.421 to 1.282; I^2^ = 19.61; τ^2^ = 0.22; NNTB = 174; 95% CI = NNTH 48 to ∞ to NNTB 31) between the palonosetron (5.1%; 28 of 550 patients) and ramosetron (5.7%; 31 of 547 patients) groups.

For subgroup analysis, no evidence of differences was found between the palonosetron and ramosetron groups when they were administered early (RR = 0.535; 95% CI = 0.278 to 1.028; I^2^ = 0.0; τ^2^ = 0.0) and late (RR = 1.722; 95% CI = 0.589 to 5.036; I^2^ = 19.66; τ^2^ = 0.46; Figure 4A).

Only 4.7% (1097 of 23,261 patients) of the RIS was accrued according to the TSA. Due to limited information, the trial sequential monitoring boundary was ignored. The cumulative Z curve did not cross and stayed inside the conventional test boundary (Appendix A).

Ten studies (1097 patients) reported the incidence of late POV [26,30,31,32,33,35,37,38,40,42]. The incidence of late POV was significantly lower (RR = 0.604; 95% CI = 0.404 to 0.903; I^2^ = 0.0; τ^2^ = 0.0; NNTB = 27; 95% CI = NNTB 15 to NNTB 163) in the palonosetron group (5.5%; 30 of 550 patients) than in the ramosetron group (9.1%; 50 of 547 patients). For subgroup analysis, no evidence of differences was observed when they were administered early (RR = 0.604; 95% CI = 0.361 to 1.012; I^2^ = 0.0; τ^2^ = 0.0) and late (RR = 0.614; 95% CI = 0.316 to 1.193; I^2^ = 0.0; τ^2^ = 0.0; Figure 4B).

Only 15.4% (1097 of 7127 patients) of the RIS was accrued in the TSA. The cumulative Z curve crossed the conventional test border; however, it did not cross the trial sequential monitoring boundary (Appendix A).

Fourteen studies (1359 patients) reported the incidence of overall POV [26,27,28,30,31,32,33,34,35,37,38,39,40,42]. The combined results showed no evidence of a difference (RR = 0.879; 95% CI = 0.624 to 1.240; I^2^ = 38.77; τ^2^ = 0.30; NNTB: 111; 95% CI = NNTH 43 to ∞ to NNTB 24) between the palonosetron (9.7%; 66 of 680 patients) and ramosetron (10.6%; 72 of 679 patients) groups.

For subgroup analysis, the incidence of the overall POV was significantly lower in the palonosetron group than in the ramosetron group when they were administered early (RR = 0.523; 95% CI = 0.311 to 0.880; I^2^ = 0.0; τ^2^ = 0.0), but no evidence of difference was observed between the palonosetron and ramosetron groups when they were administered late (RR = 1.029; 95% CI = 0.525 to 2.018; I^2^ = 47.42; τ^2^ = 0.82) and at different time points (RR = 1.623; 95% CI = 0.869 to 3.029; I^2^ = 0.0; τ^2^ = 0.0; Figure 4C).

Only 7.2% (1359 of 18,799 patients) of the RIS was accrued in the TSA. The cumulative Z curve crossed the conventional test border, however, did not cross the trial sequential monitoring boundary (Appendix A).

### 3.5. PONV (Early, Late, and Overall Phase)

Nine studies (1200 patients) reported the incidence of early PONV [29,31,33,34,36,38,39,40,41]. The combined results showed no evidence of a difference (RR = 0.984; 95% CI = 0.831 to 1.164; I^2^ = 45.57; τ^2^ = 0.07; NNTB = 413; 95% CI = NNTH 20 to ∞ to NNTB 19) between the palonosetron (27.2%; 163 of 599 patients) and ramosetron (27.5%; 165 of 601 patients) groups.

For subgroup analysis, no evidence of differences was observed between the palonosetron and ramosetron groups when they were administered early (RR = 0.707; 95% CI = 0.486 to 1.029; I^2^ = 0.0; τ^2^ = 0.0), late (RR = 1.063; 95% CI = 0.873 to 1.296; I^2^ = 71.89; τ^2^ = 0.16) and different time points (RR = 1.131; 95% CI = 0.606 to 2.110; I^2^ = 0.0; τ^2^ = 0.0; Figure 5A).

Only 19.9% (1200 of 6016 patients) of the RIS was accrued in the TSA. The cumulative Z curve did not cross the conventional test boundary (Appendix A).

Nine studies (1200 patients) reported the incidence of late PONV [29,31,33,34,36,38,39,40,41].

The combined results showed no evidence of a difference (RR = 0.983; 95% CI = 0.800 to 1.209; I^2^ = 0.0; τ^2^ = 0.0; NNTB = 432; 95% CI = NNTH 20 to ∞ to NNTB 18) between the palonosetron (30.6%; 183 of 599 patients) and ramosetron (30.8%; 185 of 601 patients) groups. For subgroup analysis, no evidence of differences was observed between the palonosetron and ramosetron groups when they were administered early (RR = 0.653; 95% CI = 0.322 to 1.322; I^2^ = 0.0; τ^2^ = 0.0), late (RR = 1.021; 95% CI = 0.819 to 1.272; I^2^ = 71.89; τ^2^ = 0.16), and at different time points (RR = 1.061; 95% CI = 0.326 to 1.209; I^2^ = 0.0; τ^2^ = 0.0; Figure 5B).

Only 23.3% (1200 of 5144 patients) of the RIS was accrued in the TSA. The cumulative Z curve did not cross the conventional test border (Appendix A).

Nine studies (1200 patients) reported the incidence of overall PONV [29,31,33,34,36,38,39,40,41]. The combined results showed no evidence of a difference (RR = 1.073; 95% CI = 0.957 to 1.203; I^2^ = 11.41; τ^2^ = 0.01; NNTB = 432; 95% CI = NNTH 20 to ∞ to NNTB 18) between the palonosetron (30.6%; 183 of 599 patients) and ramosetron (30.8%; 185 of 601 patients) groups. For subgroup analysis, no evidence of differences was observed between the palonosetron and ramosetron groups when they were administered early (RR = 0.956; 95% CI = 0.743 to 1.230; I^2^ = 0.0; τ^2^ = 0.0), late (RR = 1.102; 95% CI = 0.948 to 1.282; I^2^ = 59.34; τ^2^ = 0.05), and at different time points (RR = 1.113; 95% CI = 0.874 to 1.418; I^2^ = 0.0; τ^2^ = 0.0; Figure 5C).

Only 46.0% (1200 of 2607 patients) of the RIS was accrued in the TSA. The cumulative Z curve did not cross the conventional test border (Appendix A).

### 3.6. Retching, Complete Response, and Use of Rescue Antiemetics

Seven studies (592 patients) reported the retching [26,28,32,33,34,35,42]. The incidence of retching was significantly lower (RR = 0.525; 95% CI = 0.390 to 0.707; I^2^ = 0.0; τ^2^ = 0.0; NNTB: 7; 95% CI = NNTB 5 to NNTB 13) in the palonosetron group (14.5%; 43 of 296 patients) than in the ramosetron group (29.1%; 86 of 296 patients). For subgroup analysis, the incidences of retching were significantly lower in the palonosetron group than in the ramosetron group when they were administered early (RR = 0.550; 95% CI = 0.360 to 0.840; I^2^ = 0.0; τ^2^ = 0.0) and late (RR = 0.320; 95% CI = 0.148 to 0.692; I^2^ = 25.75; τ^2^ = 0.16; Figure 6A).

Only 33.0% (592 of 1792 patients) of the RIS was accrued in the TSA. The cumulative Z curve crossed the conventional test border and the trial sequential monitoring boundary (complete red curve; Appendix A).

Seven studies (547 patients) reported the complete response [26,30,32,36,37,39,42].

The combined results showed no evidence of a difference (RR = 1.028; 95% CI = 0.961 to 1.100; I^2^ = 65.12; τ^2^ = 0.01; NNTH = 44; 95% CI = NNTH 11 to ∞ to NNTB 22) between the palonosetron (81.0%; 222 of 274 patients) and ramosetron (78.8%; 215 of 273 patients) groups. For subgroup analysis, no evidence of differences was observed between the palonosetron and ramosetron groups when they were administered early (RR = 1.028; 95% CI = 0.960 to 1.101; I^2^ = 0.0; τ^2^ = 0.0) and late (RR = 1.027; 95% CI = 0.687 to 1.538; I^2^ = 82.46; τ^2^ = 0.13; Figure 6B).

Only 48.4% (547 of 1130 patients) of the RIS was accrued in the TSA. The cumulative Z curve crossed the futility boundary (Appendix A).

Thirteen studies (1448 patients) reported the use of rescue antiemetics [26,28,30,31,32,33,34,36,37,38,39,41,42]. The combined results showed no evidence of a difference (RR = 0.815; 95% CI = 0.617 to 1.077; I^2^ = 56.55; τ^2^ = 0.14; NNTB = 111; 95% CI = NNTH 43 to ∞ to NNTB 24) between the palonosetron (9.7%; 66 of 680 patients) and ramosetron (10.6%; 72 of 679 patients) groups.

For subgroup analysis, the incidence of rescue antiemetics was significantly lower in the palonosetron group than in the ramosetron group when they were administered early (RR = 0.642; 95% CI = 0.451 to 0.914; I^2^ = 5.6; τ^2^ = 0.0), but no evidence of difference was observed between the palonosetron and ramosetron groups when they were administered late (RR = 1.296; 95% CI = 0.717 to 2.342; I^2^ = 67.08; τ^2^ = 0.21) and at different time points (RR = 1.084; 95% CI = 0.542 to 2.172; I^2^ = 75.52; τ^2^ = 0.19; Figure 6C).

Only 19.4% (1448 of 7458 patients) of the RIS was accrued in the TSA. The cumulative Z curve did not cross the conventional test boundary (Appendix A).

### 3.7. Adverse Effects

Ten studies (1355 patients) reported dizziness [26,28,29,31,37,38,39,40,41,42]. The combined results showed no evidence of differences (RR = 0.933; 95% CI = 0.752 to 1.156; I^2^ = 0.0; τ^2^ = 0.0; NNTB = 124; 95% CI = NNTH 31 to ∞ to NNTB 21) between the palonosetron (16.9%; 114 of 676 patients) and ramosetron (17.7%; 120 of 679 patients) groups. For subgroup analysis, no evidence of differences was observed between the palonosetron and ramosetron groups when they were administered early (RR = 0.886; 95% CI = 0.429 to 1.849; I^2^ = 35.23; τ^2^ = 0.31) and late (RR = 1.158; 95% CI = 0.786 to 1.706; I^2^ = 0.0; τ^2^ = 0.0; Appendix A).

Only 40.4% (1355 of 3356 patients) of the RIS was accrued in the TSA. The cumulative Z curve did not cross the conventional test border (Appendix A).

Eleven studies (1413 patients) reported the headache [26,28,29,31,34,37,38,39,40,41,42]. The combined results showed no evidence of a difference (RR = 1.217; 95% CI = 0.892 to 1.660; I^2^ = 0.0; τ^2^ = 0.0; NNTB = 124; 95% CI = NNTH 31 to ∞ to NNTB 21) between the palonosetron (16.9%; 114 of 676 patients) and ramosetron (17.7%; 120 of 679 patients) groups. For subgroup analysis, no evidence of differences was found between the palonosetron and ramosetron groups when they were administered early (RR = 1.042; 95% CI = 0.589 to 1.843; I^2^ = 0.0; τ^2^ = 0.0), late (RR = 1.145; 95% CI = 0.729 to 1.799; I^2^ = 0.0; τ^2^ = 0.0), and at different time points (RR = 1.682; 95% CI = 0.881 to 3.211; I^2^ = 4.93; τ^2^ = 0.07; Appendix A).

Only 20.3% (1413 of 6960 patients) of the RIS was accrued in the TSA. The cumulative Z curve did not cross the conventional test boundary (Appendix A).

### 3.8. Publication Bias

Egger’s linear regression methods were performed for the following outcomes: early PON (Intercept = −1.090; 95% CI = −2.489 to 0.308; *p* = 0.113), early POV (Intercept = 0.933; 95% CI = −0.638 to 2.503; *p* = 0.208), late PON (Intercept = −0.877; 95% CI = −2.344 to 0.591; *p* = 0.212), late POV (Intercept = 0.072; 95% CI = −0.675 to 0.819; *p* = 0.829), overall PON (Intercept = −1.148; 95% CI = −2.312 to 0.156; *p* = 0.053), overall POV (Intercept = −0.238; 95% CI = −1.844 to 0.737; *p* = 0.753), rescue antiemetics (Intercept = −0.372; 95% CI = −2.364 to 1.620; *p* = 0.686), complete response (Intercept = 0.146; 95% CI = −0.929 to 1.220; *p* = 0.762), dizziness (Intercept = 0.146; 95% CI = −0.929 to 1.221; *p* = 0.762), and headache (Intercept = −0.705; 95% CI = −1.713 to 0.303; *p* = 0.148).

The following outcomes showed asymmetry in the funnel plot: early PON, late PON, late POV, overall PON, and headache (Figure 7). Thus, trim and fill analysis was conducted for the presence of publication bias, but there was no significant change in the results.

### 3.9. Risk of Bias

The overall risks of bias are summarized in Table 3. The information about allocation concealment was provided in seven studies [26,28,29,34,39,40,41]. Only four studies described the information of registry [29,35,36,41]. Therefore, only two studies were graded as “low risk,” [29,41] and the other studies were graded as “some concerns” or “high risk.”

### 3.10. Quality of the Evidence

Fourteen outcomes were assessed to evaluate the quality of evidence using the GRADE system. The quality of evidence for each outcome was rated as low, moderate, or high (Table 4). The evidence quality of the pooled analysis of the complete response was low, but the evidence quality of the pooled analysis of late PON, rescue antiemetics, and retching was moderate. Otherwise, the evidence quality of the pooled analysis of other outcomes was high.

## 4. Discussion

The results of the current updated systematic review and meta-analysis with trial sequential analysis showed that the prophylactic administration of palonosetron was more effective than ramosetron to prevent the development of postoperative late POV and retching. Although the superiority of palonosetron compared to ramosetron was demonstrated in late POV and retching, no evidence of differences was observed between the effectiveness of palonosetron and ramosetron in preventing PON, POV, and PONV at early, late, and overall phases (except that in late POV), use of rescue antiemetic, and complete response. Headaches or dizziness also did not show statistical differences between palonosetron and ramosetron recipients. Palonosetron was superior to ramosetron when the 5-HT_3_ receptor antagonist was administered during the early phase of the surgery, in terms of early PON, late PON, overall POV, and use of rescue antiemetics. Ramosetron, however, outperformed palonosetron when it was administered at the end of surgery in preventing early PON. In terms of retching, palonosetron was superior to ramosetron when they were administered during the early phase of the operation and at the end of surgery.

At the caudal end of the fourth ventricle, the area postrema is located on the dorsal surface of the medulla oblongata and hosts the vomiting center to regulate emesis. The development of emesis is triggered by various pathways: (1) vagal afferent fibers in the gut, (2) input from the vestibular system, (3) chemoreceptor trigger zone, and (4) the forebrain. The 5-HT3 receptor, which is distributed in the central nervous system, peripheral nervous system, and intestinal tissues, plays a physiological role in coordinating emesis [43]. 5-HT3 receptor antagonists are superior to other conventional antiemetic drugs in preventing PONV [8,44]. Among them, palonosetron and ramosetron, which are recently developed selective 5-HT3 receptor antagonists, have a well-established effect in preventing PONV; therefore, they are widely used.

Our previous meta-analyses failed to find a difference between palonosetron and ramosetron in preventing PON, POV, and PONV [7]. Since only a small number of studies and cases were included, the quality of the evidence was limited. After our previous systematic review and meta-analysis, many well-designed large-scale RCTs were performed and published. Therefore, this systematic review and meta-analysis could include 17 RCTs with a total of 1823 patients.

In the current updated systematic review and meta-analysis with trial sequential analysis, palonosetron was more effective than ramosetron in preventing late POV and retching. It is inferred that the superior effect of palonosetron for the prevention of late POV is related to the longer action duration of palonosetron. Palonosetron has a longer half-life of approximately 40 h than that of ramosetron, which is 5.8 h [45,46]. Retching is defined as the expulsive movement of the esophagus and stomach muscles without the expulsion of gastric contents and is considered an emetic episode with vomiting [47]. In TSA, the cumulative *Z* curve did not cross the trial sequential boundary in preventing late POV but crossed the trial sequential monitoring boundary in preventing retching. This finding suggested the results of TSA reached a sufficient evidence level; therefore, firm conclusions about the superior effectiveness of palonosetron to ramosetron in preventing retching could be drawn.

A subgroup analysis demonstrated significant differences in the effectiveness of the 5-HT3 receptor antagonist depending on administration time. Palonosetron was superior to ramosetron when it was administered at the early phase of surgery in terms of early PON, late PON, overall POV, and use of rescue antiemetics. Conversely, ramosetron was superior to palonosetron when it was administered at the end of surgery with respect to early PON. This finding supported the results of our previous meta-analysis, which revealed the different levels of effectiveness of palonosetron and ramosetron depending on administration time [7]. Ramosetron has been reported in numerous studies supporting the benefits of administration at the end of surgery and is recommended at an A2 level evidence [4]. In the case of palonosetron, however, no recommendations on administration time were provided in the guideline. Our result suggested the benefits of the early administration of palonosetron. Exceptionally, palonosetron was proven to be more effective in preventing retching regardless of administration timing. The superior effect of the early administration of palonosetron is likely related to its long action duration. It has a longer terminal elimination half-life of approximately 40 h than that of ramosetron, which is 5.8 h [45,46].

Our study has several limitations. First, significant differences in palonosetron and ramosetron based on administration time were identified in conventional meta-analysis, but no significant differences were shown in the TSA. Nonetheless, the incidence of retching, which is part of an emetic episode, is significantly lower in the palonosetron group than in the ramosetron group. Second, only the published trials were included in this meta-analysis. Nonetheless, our current meta-analysis is a systematic review encompassing the maximum number of trials and involving TSA in the methodology to compare the effectiveness of palonosetron and ramosetron in the PONV prevention. Lastly, the cumulative Z curve of late POV did not cross the trial sequential monitoring boundary, even though it crossed the conventional test boundary. Therefore, further research is required as a definite conclusion cannot be derived. Despite these limitations, our results of the systematic review and meta-analysis with TSA provided convincing evidence to support the superior effect of palonosetron in preventing the emetic episode of vomiting.

## 5. Conclusions

In conclusion, the prophylactic administration of palonosetron was more effective in preventing the development of retching postoperatively. Subgroup analysis showed that palonosetron was more effective than ramosetron against retching regardless of the timing of administration. The prophylactic administration of palonosetron and ramosetron did not show significant differences in the incidence of PON, POV, and PONV. Subgroup analysis identified that palonosetron was more effective when it was administered early in surgery, whereas ramosetron was more effective when it was administered late in surgery.

## Figures and Tables

**Figure 1 jpm-13-00082-f001:**
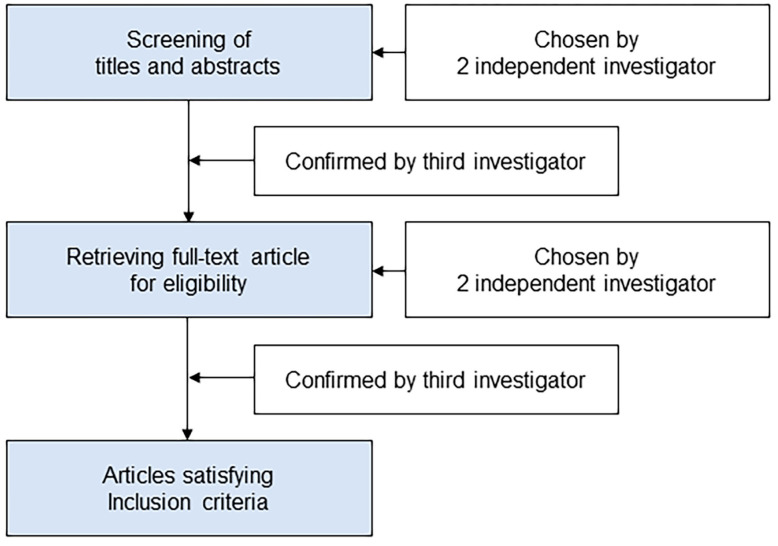
Schematic flow diagram of the study selection.

**Figure 2 jpm-13-00082-f002:**
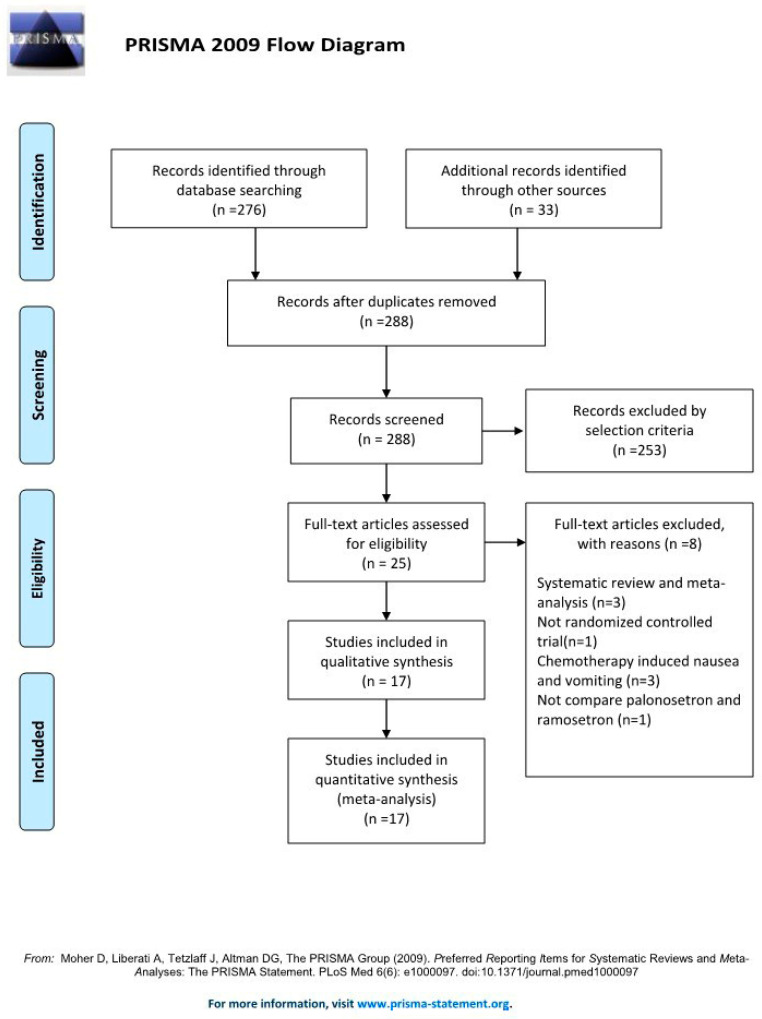
PRISMA flow diagram of the search for randomized controlled trials and the inclusion and exclusion criteria [10].

**Figure 3 jpm-13-00082-f003:**
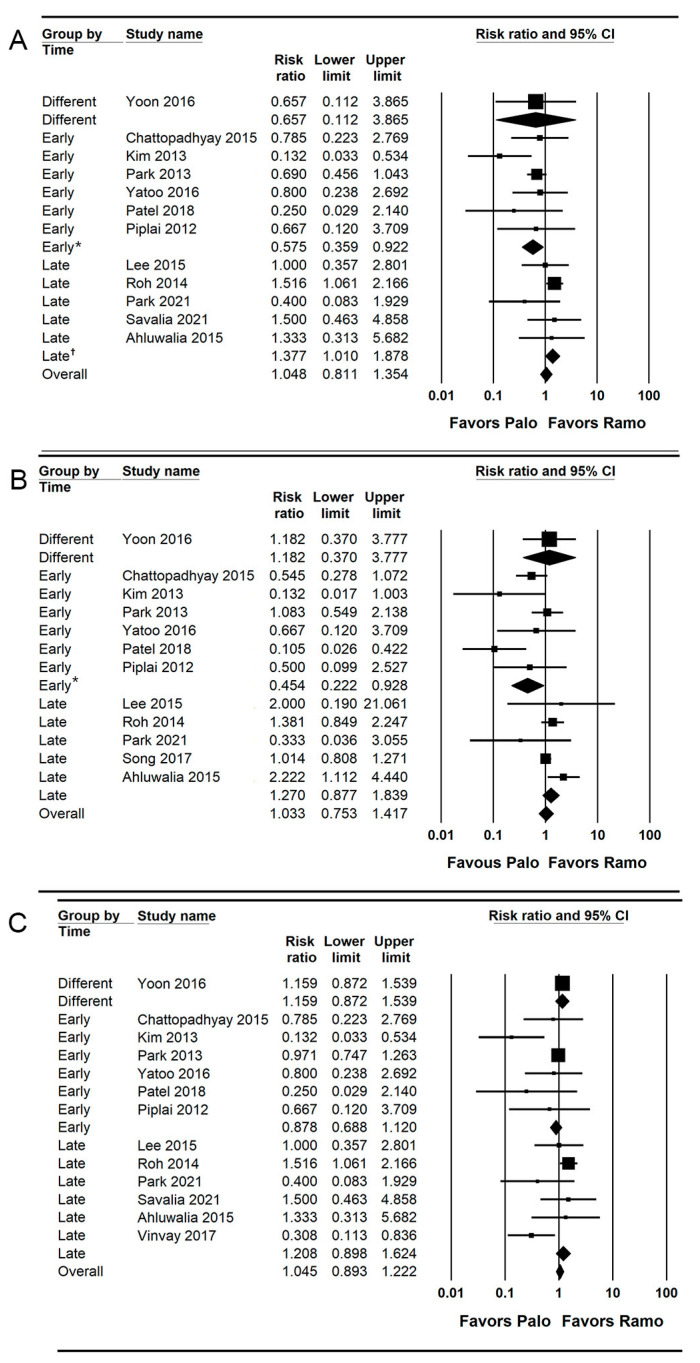
Forest plot of studies comparing the effectiveness of palonosetron with that of ramosetron on PON. (**A**) Early, (**B**) late, and (**C**) overall. The size of the filled squares of risk ratios reflects the effect size of individual trials. Horizontal bars represent 95% confidence intervals of difference. The diamond shape depicted the pooled estimates and uncertainty of the combined effects on early, late, and overall PON. The combined results showed no evidence of differences between palonosetron and ramosetron. For subgroup analysis, the pooled estimates of palonosetron showed that it was more effective than ramosetron in preventing early and late PON when they were administered early (*). Conversely, the pooled estimates of ramosetron indicated that it was more effective than palonosetron in preventing early PON when they were administered late (†).

**Figure 4 jpm-13-00082-f004:**
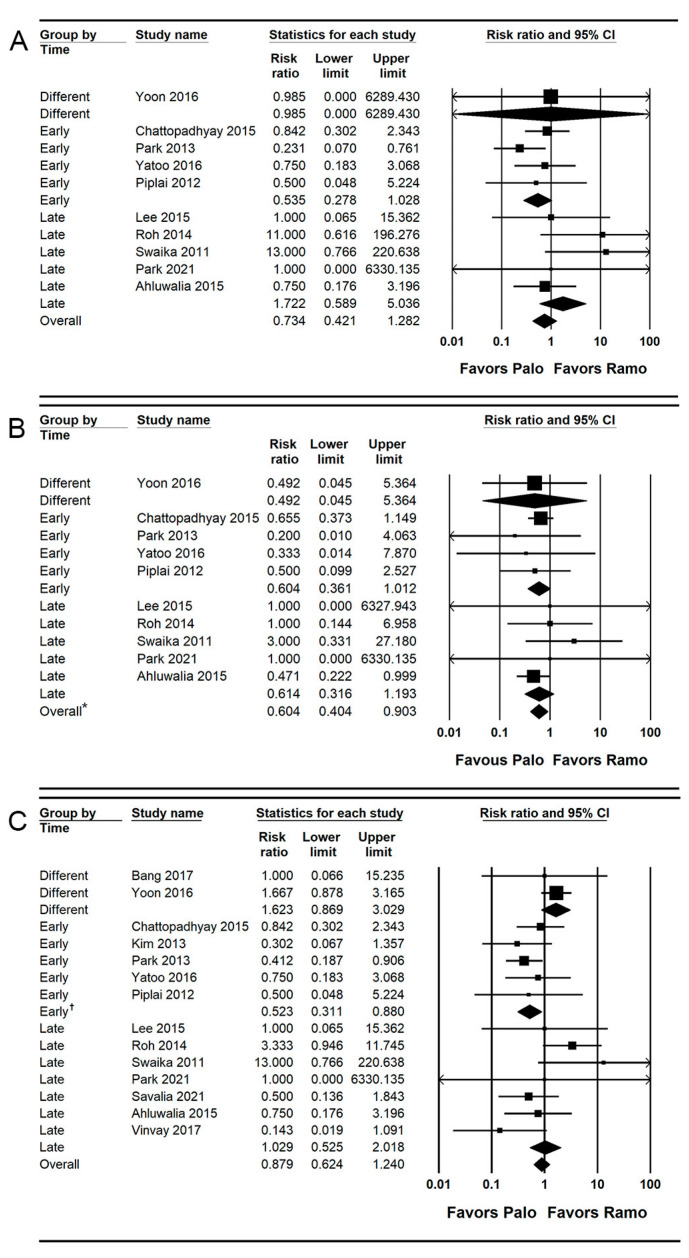
Forest plot of studies comparing the effectiveness of palonosetron with that of ramosetron on POV. (**A**) Early, (**B**) late, and (**C**) overall. The size of the filled squares of risk ratios reflects the effect size of individual trials. Horizontal bars represent 95% confidence intervals of difference. The diamond depicted the pooled estimates and uncertainty of the combined effect. The combined results showed no evidence of differences between palonosetron and ramosetron for early and overall POV. Notwithstanding, palonosetron was more effective than ramosetron in preventing late POV (*). For subgroup analysis, the pooled estimates of palonosetron indicated that it was more effective than ramosetron in preventing the overall POV when they were administered early (†), while the pooled estimates presented no evidence of differences between palonosetron and ramosetron when they were administered late.

**Figure 5 jpm-13-00082-f005:**
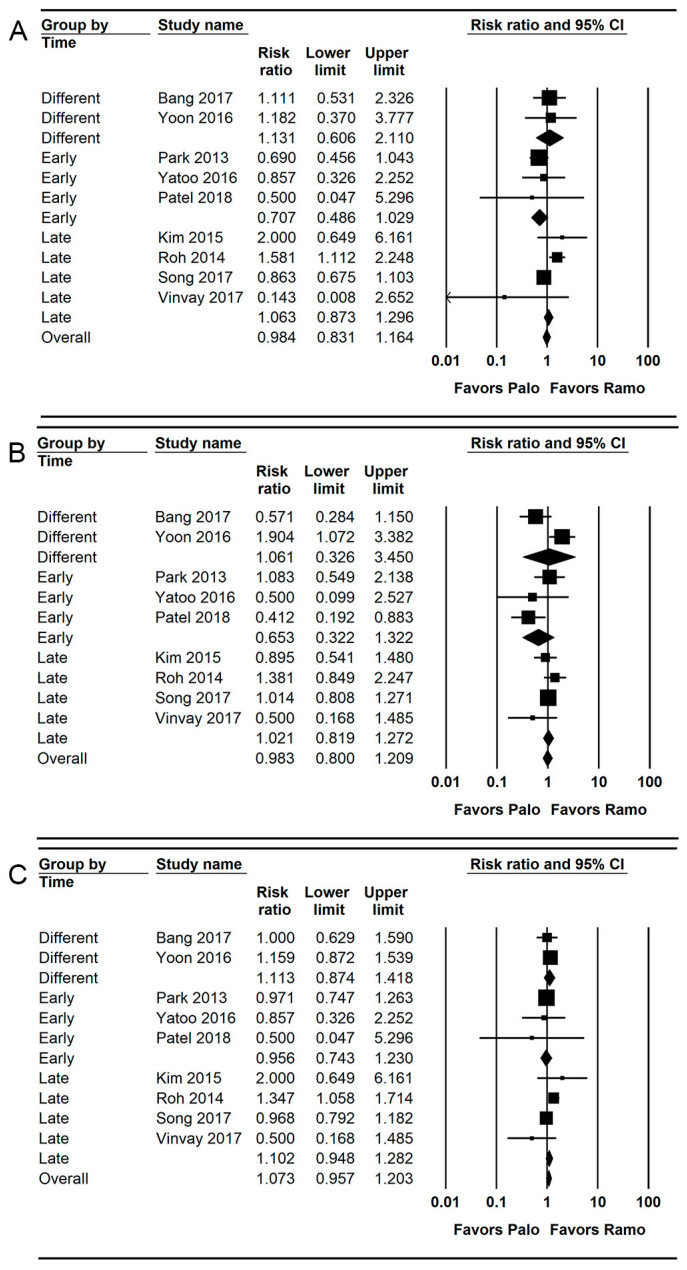
Forest plot of studies comparing the effectiveness of palonosetron with that of ramosetron on PONV. (**A**) Early, (**B**) late, and (**C**) overall. The size of the filled squares of risk ratios reflects the effect size of individual trials. Horizontal bars represent 95% confidence intervals of difference. The diamond depicted the pooled estimates and uncertainty of the combined effect. The combined results showed no evidence of differences between palonosetron and ramosetron for early, late, and overall PONV. For subgroup analysis, no evidence of differences was observed between palonosetron and ramosetron according to administration time.

**Figure 6 jpm-13-00082-f006:**
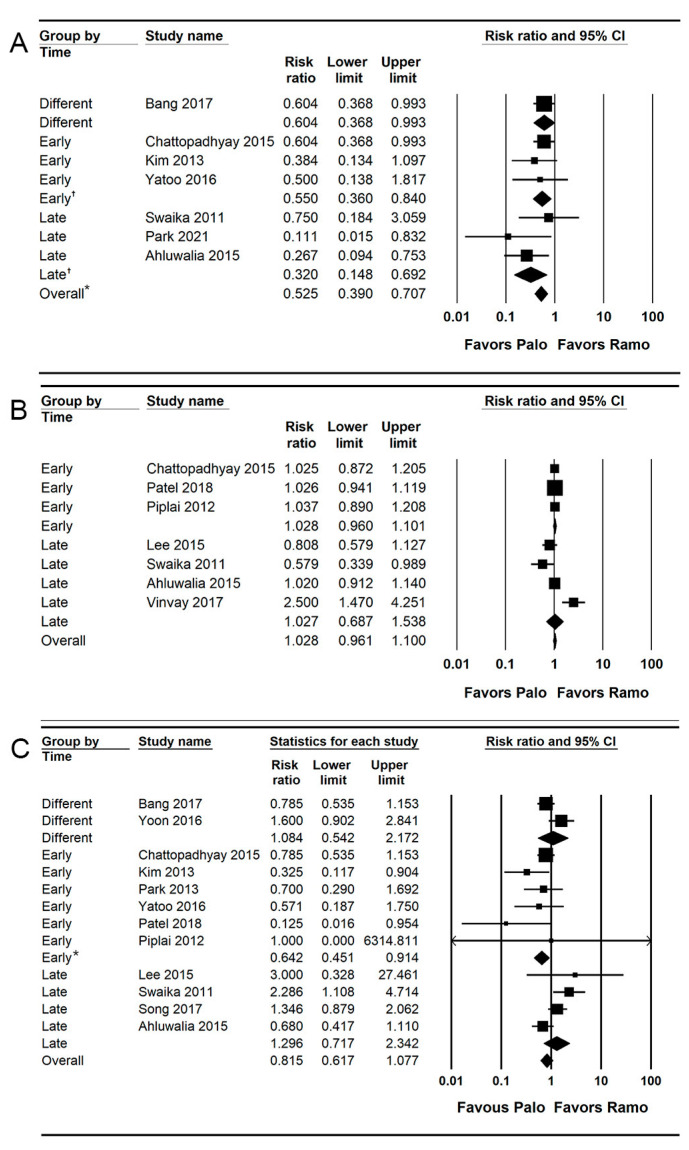
Forest plot of studies comparing the effectiveness of palonosetron with that of ramosetron on (**A**) retching, (**B**) complete response, and (**C**) use of rescue antiemetics. The diamond shape depicted the pooled estimates and uncertainty of the combined effect. The combined results showed no evidence of differences between palonosetron and ramosetron for the complete response and use of rescue antiemetics. On the other hand, palonosetron was more effective than ramosetron in preventing retching (*). For subgroup analysis, no evidence of differences was observed between palonosetron and ramosetron for the complete response according to administration time. Whereas, the pooled estimates demonstrated that palonosetron was more effective than ramosetron in the prevention of retching when they were administered early and late (†) and in the use of rescue antiemetics when they were administered early (*).

**Figure 7 jpm-13-00082-f007:**
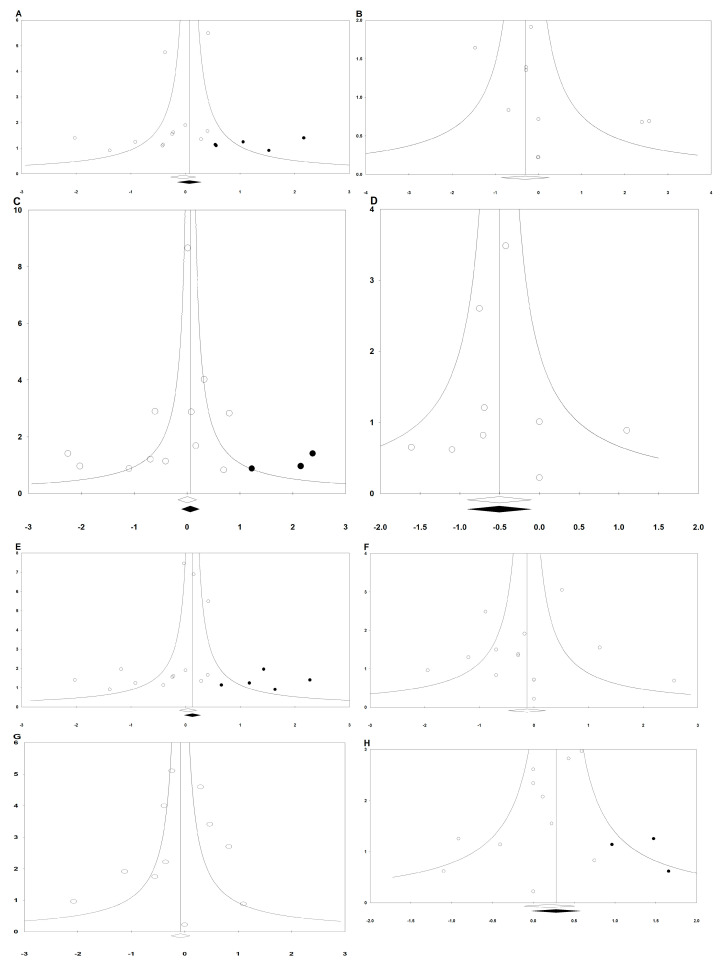
Funnel plots of (**A**) early PON, (**B**) early POV, (**C**) late PON, (**D**) late POV, (**E**) overall PON, (**F**) overall POV, (**G**) rescue antiemetics, and (**H**) headache. The horizontal axis demonstrates the log of risk ratio, and the vertical axis shows the inverse of the standard error.

**Table 1 jpm-13-00082-t001:** Study characteristics of included studies.

Source	Age (yrs)	Sex (M/F)	Weight (kg)	Height (cm)	ASA-PS	Risk Factors for PONV	Type of Anesthesia	Duration of Anesthesia (min)	Type of Surgery
Chattopadhyay 2015 [26]	18–35	0/109	58.8 [7.2]	NR	I-II	≥2 (F, nonsmoking)	SA	60.5 (4.1)	Elective cesarean delivery
Kim 2013 [28]	20–65	0/74	65 [1.3]	164.5 [4.9]	I-II	≥3 (F, IV-PCA, nonsmoking)	GA	169.39 (87.6)	Laparoscopic surgery
Kim 2015 [29]	NR	0/88	59 [9]	158 [5]	NR	≥2 (F, nonsmoking)	GA	146 (44)	Gynecologic laparoscopic surgery
Lee 2015 [30]	NR	0/70	60.1 [4.9]	155.3 [3.1]	I-II	≥1 (F)	GA	128.1 (47.5)	Laparoscopic hysterectomy
Park 2013 [31]	≥20	0/100	61.8 [8.5]	158.9 [5.8]	I-II	≥1 (IV-PCA)	GA	143.4 (53.8)	Gynecologic laparoscopic surgery
Roh 2014 [40]	20–65	0/196	NR	NR	NR	≥1 (IV-PCA)	GA	168 (66)	Lumbar spinal surgery
Swaika 2011 [32]	18–70	0/58	52.8 [6.9]	NR	I-II	≥1 (F)	GA	56.1 (8.0)	Laparoscopic cholecystectomy
Yatoo 2016 [33]	18–65	31/29	65.4 [4.8]	157.4 [7.2]	I-II	≥0	GA	42.6 (9.4)	Elective laparoscopic surgery
Bang 2017 [34]	20–49	0/87	59.21 [9.54]	159.02 [4.97]	I-II	≥3 (F, nonsmoker, IV-PCA)	GA	40 (9.6)	Gynecologic laparoscopic surgery
Park 2021 [35]	20–70	0/108	58.6 [10.6]	160.6 [5.9]	I-II	≥2 (F, IV-PCA)	GA	40.7 (11.2)	Gynecologic laparoscopic surgery
Patel 2018 [36]	18–60	NR	NR	NR	I-II	≥0	GA	NR	Laparoscopic surgery
Piplai 2012 [37]	18–65	NR	54.4 [8.22]	NR	I-II	≥1 (F)	GA	43.4 (6.46)	Laparoscopic cholecystectomy
Savalia 2021 [27]	18–60	NR	NR	NR	I-II	≥0	GA	NR	Middle ear surgery
Yoon 2016 [38]	≥20	0/262	60.1 [7.1]	158.2 [5.7]	I-II	≥2 (F, IV-PCA)	GA	51.2 (11.1)	Oncologic gynecologic surgery
Song 2017 [41]	20–85	NR	NR	NR	I-II	≥0	GA	55 (14)	Spinal surgery
Ahluwalia 2015 [42]	25–40	0/60	54.52 [5.21]	154.27 [2.87]	I-II	≥1 (F)	GA	133.62 (9.83)	Middle ear surgery
Vinvay 2017 [39]	20–50	46/14	58.63 [9.99]	155.53 [8.28]	I-II	≥0	GA	NR	Elective laparoscopic surgery and ENT surgery

yrs; years, M; male, F; female, ASA-PS; American Society of Anesthesiologists Physical Status, PONV; postoperative nausea and vomiting, min; minutes, NR; not reported, GA; general anesthesia, SA; spinal anesthesia, IV-PCA; intravenous patient-controlled anesthesia, GA; general anesthesia, ENT; ear, nose and throat. Values of weight, height, and duration of anesthesia are expressed mean (standard deviation).

**Table 2 jpm-13-00082-t002:** Further study characteristics of included studies.

Source	Data Collection Period	Dose of Palonosetron/Ramosetron	Administration Timing	Rescue Antiemetics
Chattopadhyay 2015 [26]	0–2/2–24/24–48 h	0.075 mg/0.3 mg	Immediate after clamping of the fetal umbilical cord	Metoclopramide 10 mg
Kim 2013 [28]	0–1/1–6/6–24/24/48 h	0.075 mg/0.3 mg	Just prior to induction of anesthesia	1st choice, propofol 20 mg, metoclopramide 10 mg; 2nd choice, ondansetron 4 mg or/and dexamethasone 4 mg
Kim 2015 [29]	Arrival PACU/ discharge PACU/ 24 h/48 h/72 h	0.075 mg/0.3 mg	10 min at the end of operation	Metoclopramide 10 mg
Lee 2015 [30]	0–6/6–24/24–48 h	0.075 mg/0.3 mg	At the end of the operation, prior to extubation	Metoclopramide 10 mg
Park 2013 [31]	0–6/6–24/24–48 h	0.075 mg/0.3 mg	Immediately before the induction of anesthesia	Metoclopramide 10 mg
Roh 2014 [40]	PACU/0–6/6–24/24–48/48–72 h	0.075 mg/0.3 mg	Immediately before the induction of anesthesia	Metoclopramide 10 mg
Swaika 2011 [32]	0–2/2–6/6–24 h	0.075 mg/0.3 mg	Just at the end of operation before extubation	Ondansetron 4 mg
Yatoo 2016 [33]	0–4/4–12/24–48 h	0.075 mg/0.3 mg	5 min before the induction	Metoclopramide 0.15 mg/kg
Bang 2017 [34]	0–2 h/2–48 h /30 min/60 min/ 90 min/120 min/6 h/48 h	0.075 mg/0.3 mg	Palonosetron: immediately before anesthesia induction Ramosetron: 30 min before the end of operation	1st choice, Ondansetron 4 mg; 2nd choice Ramosetron 0.3 mg
Park 2021 [35]	PACU/0–6/6–24/24–48 h	0.075 mg/0.3 mg	Mixed with IV-PCA	Metoclopramide 10 mg
Patel 2018 [36]	0–6/6–24/24–72 h	0.075 mg/0.3 mg	Just before induction of anesthesia	Ondansetron 4 mg
Piplai 2012 [37]	0–3/3–24/24–48/48–72 h	0.075 mg/0.3 mg	Before induction of anesthesia	Metoclopramide 10 mg
Savalia 2021 [27]	0–6/6–12/12–24/24–48 h	0.075 mg/0.3 mg	Before the end of operation	NR
Yoon 2016 [38]	0–3/3–24/24–48 h	0.075 mg/0.3 mg	Palonosetron: immediately after anesthesia induction Ramosetron: 30 min before the end of operation	Metoclopramide 10 mg
Song 2017 [41]	0–6/6–48/0–48 h	0.075 mg/0.3 mg	20 min before the end of operation and 24 h after operation	Metoclopramide 10 mg
Ahluwalia 2015 [42]	0–2/2–24/24–48 h	0.075 mg/0.3 mg	Before shifting the patient from operation room to PACU	Metoclopramide 10 mg
Vinay 2017 [39]	0–4/5–12/12 h-overall	0.075 mg/0.3 mg	Before shifting the patient from operation room to PACU	Metoclopramide 10 mg

PACU; post-anesthesia care unit, IV-PCA; intravenous patient-controlled anesthesia.

**Table 3 jpm-13-00082-t003:** Risk of bias 2.0.

Source	Bias Arising from the Randomization Process	Bias Due to Deviations from the Intended Interventions	Bias Due to Missing Outcome Data	Bias in Measurement of the Outcome	Bias in Selection of the Reported Result	Overall Risk of Bias
Chattopadhyay 2015 [26]	Low	Low	Low	Low	Some concern	Some concern
Kim 2013 [28]	Low	Low	Low	Low	Some concern	Some concern
Kim 2015 [29]	Low	Low	Low	Low	Low	Low
Lee 2015 [30]	Some concern	Low	Low	Low	Some concern	High
Park 2013 [31]	Some concern	Low	Low	Low	Some concern	High
Roh 2014 [40]	Low	Low	Low	Low	Some concern	Some concern
Swaika 2011 [32]	Some concern	Low	Low	Some concern	Some concern	High
Yatoo 2016 [33]	Some concern	Low	Some concern	Some concern	Some concern	High
Bang 2017 [34]	Low	Low	Low	Low	Some concern	Some concern
Park 2021 [35]	Some concern	Low	Low	Low	Low	Some concern
Patel 2018 [36]	Some concern	Low	Low	Low	Low	Some concern
Piplai 2012 [37]	Some concern	Low	Low	Low	Some concern	High
Savalia 2021 [27]	Some concern	Low	Some concern	Some concern	Some concern	High
Yoon 2016 [38]	Some concern	Low	Low	Some concern	Some concern	High
Song 2017 [41]	Low	Low	Low	Low	Low	Low
Ahluwalia 2015 [42]	Some concern	Low	Low	Low	Some concern	High
Vinay 2017 [39]	Low	Low	Low	Some concern	Some concern	High

**Table 4 jpm-13-00082-t004:** The results of meta-analysis and GRADE evidence quality for each outcome.

	No of Studies	No of Patients	Conventional Meta-Analysis			Trial Sequential Analysis			NNT	Quality Assessment					Quality
			RR with 95% CI	Heterogeneity (I^2^)	Publication Bias (Egger’s Test)	Conventional Test Boundary	Monitoring Boundary	RIS		ROB	Inconsistency	Indirectness	Imprecision	Publication Bias	
Early PON	12	1263	RR: 1.048; 95% CI 0.811 to 1.354	46.34	−1.090; 95% CI −2.489 to 0.308	Not cross	Not cross	9.5% (1263 of 13,314 patients)	NNTH: 741; 95% CI NNTH 25 to ∞ to NNTB	Not serious	Not serious	Not serious	Not serious	Not serious	⨁⨁⨁⨁ High
Early POV	10	1097	RR: 0.734; 95% CI 0.421 to 1.282	19.61	0.933; 95% CI −0.638 to 2.503	Not cross	Not cross	4.7% (1097 of 23,261 patients)	NNTB: 174; 95% CI NNTH 48 to ∞ to NNTB 31	Not serious	Not serious	Not serious	Not serious	Not serious	⨁⨁⨁⨁ High
Early PONV	9	1200	RR: 0.984; 95% CI 0.831 to 1.164	45.57	NA	Not cross	Not cross	19.9% (1200 of 6016 patients)	NNTB: 413; 95% CI NNTH 20 to ∞ to NNTB 19	Not serious	Not serious	Not serious	Not serious	NA	⨁⨁⨁⨁ High
Late PON	12	1489	RR: 1.033; 95% CI 0.753 to 1.417	57.93	−0.877; 95% CI −2.344 to 0.591	Not cross	Not cross	10.1% (1489 of 14,733 patients)	NNTB: 1911; 95% CI NNTH 25 to ∞ to NNTB 24	Not serious	Serious	Not serious	Not serious	Not serious	⨁⨁⨁◯ Moderate
Late POV	10	1097	RR: 0.604; 95% CI 0.404 to 0.903	0.0	0.072; 95% CI −0.675 to 0.819	Cross	Not cross	15.4% (1097 of 7127 patients)	NNTB: 27; 95% CI NNTB 15 to NNTB 163	Not serious	Not serious	Not serious	Not serious	Not serious	⨁⨁⨁⨁ High
Late PONV	9	1200	RR: 0.983; 95% CI 0.800 to 1.209	0.0	NA	Not cross	Not cross	23.3% (1200 of 5144 patients)	NNTB: 432; 95% CI NNTH 20 to ∞ to NNTB 18	Not serious	Not serious	Not serious	Not serious	NA	⨁⨁⨁⨁ High
Overall PON	13	1323	RR: 1.045; 95% CI 0.893 to 1.222	48.75	−1.148; 95% CI −2.312 to 0.156	Not cross	Not cross	15.2% (1323 of 8707 patients)	NNTH: 68; 95% CI NNTH 16 to ∞ to NNTB 31	Not serious	Not serious	Not serious	Not serious	Not serious	⨁⨁⨁⨁ High
Overall POV	14	1359	RR: 0.879; 95% CI 0.624 to 1.240	38.77	−0.238; 95% CI −1.844 to 0.737	Cross	Not cross	7.2% (1359 of 18,799 patients)	NNTB: 111; 95% CI NNTH 43 to ∞ to NNTB 24	Not serious	Not serious	Not serious	Not serious	Not serious	⨁⨁⨁⨁ High
Overall PONV	9	1200	RR: 1.073; 95% CI 0.957 to 1.203	11.41	NA	Not cross	Not cross	46.0% (1200 of 2607 patients)	NNTB: 432; 95% CI NNTH 20 to ∞ to NNTB 18	Not serious	Not serious	Not serious	Not serious	NA	⨁⨁⨁⨁ High
Rescue Anti-emetics	13	1448	RR: 0.815; 95% CI 0.617 to 1.077	56.55	−0.372; 95% CI −2.364 to 1.620	Not cross	Not cross	19.4% (1448 of 7458 patients)	NNTB: 111; 95% CI NNTH 43 to ∞ to NNTB 24	Not serious	Serious	Not serious	Not serious	Not serious	⨁⨁⨁◯ Moderate
Retching	7	592	RR: 0.525; 95% CI 0.390 to 0.707	0.0		Cross	Cross	33.0% (592 of 1792 patients)	NNTB: 7; 95% CI NNTB 5 to NNTB 13	Not serious	Not serious	Not serious	Serious	NA	⨁⨁⨁◯ Moderate
Complete Response	7	547	RR: 1.028; 95% CI 0.961 to 1.100	65.12	0.146; 95% CI −0.929 to 1.220	Not cross	Not cross Cross the futility boundary border	48.4% (547 of 1130 patients)	NNTH: 44; 95% CI NNTH 11 to ∞ to NNTB 22	Not serious	Serious	Not serious	Serious	NA	⨁⨁◯◯ Low
Dizziness	10	1355	RR: 0.933; 95% CI 0.752 to 1.156	0.0	0.146; 95% CI −0.929 to 1.221	Not cross	Not cross	40.4% (1355 of 3356 patients)	NNTB: 124; 95% CI NNTH 31 to ∞ to NNTB 21	Not serious	Not serious	Not serious	Not serious	Not serious	⨁⨁⨁⨁ High
Headache	11	1413	RR: 1.216; 95% CI 0.892 to 1.659	0.0	−0.705; 95% CI −1.713 to 0.303	Not cross	Not cross	40.4% (1355 of 3356 patients)	NNTB: 124; 95% CI NNTH 31 to ∞ to NNTB 21	Not serious	Not serious	Not serious	Not serious	Not serious	⨁⨁⨁⨁ High

No; number, RR; relative risk, CI; confidence interval, RIS; required information size, NNT; number needed to treat; ROB; risk of bias; PON: post-operative nausea, NNTH; number needed to treat harm, NNTB; number needed to treat benefit, POV; post-operative vomiting, PONV; post-operative nausea and vomiting, NA; not applicable.

## Data Availability

The following information was supplied regarding data availability: the raw data are available as a Appendix A.

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
