# Peer review of "Comparison of the Effectiveness of Palonosetron and Ramosetron in Preventing Postoperative Nausea and Vomiting: Updated Systematic Review and Meta-Analysis with Trial Sequential Analysis"

_jpm, 2022, doi:10.3390/jpm13010082_

Round 1

Reviewer 1 Report

In the manuscript jpm-2081251, authors aimed to compare the efficacy of the perioperative administration of palonosetron with that of ramosetron in preventing postoperative nausea and vomiting (PONV).

Generally, the paper topic is interesting than new posts as corroborating previous study/es about palonosetron administration. I suggest a minor revision because the current manuscript has several qualities that are considered adequate for publication, as follows:

  1. The manuscript is technically sound, and the data do support the conclusions.
  2. The statistical analysis has been performed appropriately.
  3. The manuscript is presented in an intelligible fashion and written in clear, correct, and unambiguous standard English. However, I have several corrections:

Lines 15-20: better to mention the number of studies with the range of studies used. Please rewrite.

Line 23-24: based on what the authors conclude in this statement? Mention earlier the factor influence/s.

Lines 47-48: Expand on your sentences; this appears to be an incomplete paragraph.

Lines 95-100: Include the year of the start of the search.

Line 119: In my view, I suggest you convert this into a scheme or still use the present sentences, but add a short scheme.

Line 202: See my previous comment about the study year.

Figures: revised with better resolution, especially figure 6.

Lines 550-552: is this your sentence/s? give the citation appropriately.

Supplementary materials: supplementary figures, except 5C, need improvement, especially in the way they depict the dot or legend.

Reviewer 2 Report

This systemic review shows that palonosetron was more effective than ramosetron to prevent the development of postoperative late POV and retching. 

Here are my enquiries.

Minor

1.  Please give any explanation on the superiority of palonosetron on retching and late POV.

2.  1st paragraph of discussion (lines 504-510): Please express more clearly that only postoperative retching and late POV showed significant difference. Please polish the sentence starting with 'However'.
